# Association between Immediate Postoperative Radiographic Findings and Failed Internal Fixation for Trochanteric Fractures: Systematic Review and Meta-Analysis

**DOI:** 10.3390/jcm11164879

**Published:** 2022-08-19

**Authors:** Norio Yamamoto, Yasushi Tsujimoto, Suguru Yokoo, Koji Demiya, Madoka Inoue, Tomoyuki Noda, Toshifumi Ozaki, Takashi Yorifuji

**Affiliations:** 1Department of Orthopedic Surgery, Miyamoto Orthopedic Hospital, Okayama 773-8236, Japan; 2Department of Epidemiology, Graduate School of Medicine, Dentistry and Pharmaceutical Sciences, Okayama University, Okayama 700-8530, Japan; 3Scientific Research Works Peer Support Group (SRWS-PSG), Osaka 541-0043, Japan; 4Departments of Health Promotion and Human Behavior, Kyoto University Graduate School of Medicine/School of Public Health, Kyoto 606-8501, Japan; 5Oku Medical Clinic, Osaka 573-0164, Japan; 6Cochrane Japan, Tokyo 104-0044, Japan; 7Department of Orthopedic Surgery, National Hospital Organization Okayama Medical Center, Okayama 701-1192, Japan; 8Department of Orthopedic Surgery, Tsuyama Chuo Hospital, Okayama 708-0841, Japan; 9Department of Orthopedic Surgery and Traumatology, Kawasaki Medical School General Medical Center, Okayama 700-8505, Japan; 10Department of Orthopedic Surgery, Okayama University Graduate School of Medicine, Dentistry and Pharmaceutical Science, Okayama 700-8530, Japan

**Keywords:** trochanteric fracture, intertrochanteric fracture, pertrochanteric fracture, reduction, tip–apex distance, mechanical failure, fixation failure, cut-out, systematic review, meta-analysis

## Abstract

Failed internal fixations for trochanteric fractures have a strong negative impact owing to increased postoperative mortality and high medical costs. However, evidence on the prognostic value of postoperative radiographic findings for failed internal fixations is limited. We aimed to clarify the association between comprehensive immediate postoperative radiographic findings and failed internal fixation using relative and absolute risk measures. We followed the meta-analysis of observational studies in epidemiology guidelines and the Cochrane handbook. We searched specific databases in November 2021. The outcomes of interest were failed internal fixation and cut-out. We pooled the odds ratios and 95% confidence intervals using a random-effects model and calculated the number needed to harm for each outcome. Thirty-six studies involving 8938 patients were included. The certainty of evidence in the association between postoperative radiographic findings and failed internal fixation or cut-out was mainly low or very low except for the association between intramedullary malreduction on the anteromedial cortex and failed internal fixation. Moderate certainty of evidence supported that intramedullary malreduction on the anteromedial cortex was associated with failed internal fixation. Most postoperative radiographic findings on immediate postoperative radiographs for trochanteric fractures were uncertain as prognostic factors for failed internal fixations.

## 1. Introduction

The incidence of geriatric trochanteric fractures has increased with the aging population [1]. Postoperative failed internal fixations leading to reoperation, such as cut-out, nonunion, and implant failure, have a strong negative impact owing to increased postoperative mortality and high medical costs [2]. In 1980, the five major factors related to the treatment outcomes following internal fixation for trochanteric fractures were: bone quality, fragment geometry, implant selection, reduction quality, and screw placement in the femoral head [3]. Among them, surgically modifiable factors are reduction quality and screw placement in the femoral head, which can be evaluated using postoperative radiographic findings.

However, evidence on the prognostic value of postoperative radiographic findings for failed internal fixations is scarce. Some postoperative radiographic findings, such as the Baumgaertner reduction criteria and a high tip–apex distance (TAD), have been investigated for their association with failed internal fixation; however, no comprehensive summary is available [4,5,6]. Therefore, the factor that has the greatest impact on failed internal fixations remains undetermined.

The aim of this systematic review and meta-analysis was to clarify the association of between all immediate postoperative radiographic findings and failed internal fixation for internally fixed trochanteric fractures in terms of relative and absolute risk measures. Identifying the relevant surgical factors and recognizing the degree of association will facilitate intraoperative decisions and postoperative follow-up planning and improve clinical outcomes.

## 2. Materials and Methods

We followed the meta-analysis of observational studies in epidemiology (MOOSE) guidelines, the Cochrane handbook [7,8], and the grading of recommendations, assessment, development, and evaluation (GRADE) criteria [9]. Prior to study initiation, we made our study protocol available in the open science framework [10].

### 2.1. Eligibility Criteria

#### 2.1.1. Study Type

We included all observational studies and secondary analyses of randomized trials investigating the association between immediate postoperative radiographic findings and failed internal fixation in trochanteric fractures. We did not apply restrictions on publication date and status (full publication, conference abstract, and unpublished data) and languages. We excluded case reports and cadaveric studies.

#### 2.1.2. Study Participants

We included patients with trochanteric fractures that were treated using internal fixation (sliding hip screw (SHS) or cephalomedullary nail (CMN)). Inclusion criteria involved the following patients: (1) aged >50 years, (2) with fracture type classified under the Arbeitsgemeinschaft für Osteosynthesefragen/Orthopedic Trauma Association (AO/OTA) classification 31A [11], and (3) with follow-up periods of more than 3 months postoperatively. The exclusion criteria were pathological fractures caused by specific pathologies other than osteoporosis, open fractures with history of operation at the ipsilateral proximal femur, and additional augmentation in fixation (cement augmentation, additional screw insertion separately from the original implant).

### 2.2. Exposures

The exposures of interest were the following five immediate postoperative radiographic findings using plain radiography: Baumgaertner reduction criteria, overall alignment in the anteroposterior (AP) radiograph, local alignment on the anteromedial cortex, TAD, and screw placement in the femoral head. We selected these radiographic findings based on earlier reviews [4,5,6]. These were evaluated using AP and/or lateral plain radiography.

### 2.3. Reduction Quality

1.The Baumgaertner reduction criteria: The reduction quality is classified as good, acceptable, or poor [12]. Poor reduction indicates varus on the AP radiograph and angulation greater than 20 degrees on the lateral radiograph and displacement more than 4 mm on both views. Here, we classified it into two categories: not poor (good or acceptable) or poor.2.Overall alignment using AP radiography: We classified it as either varus or nonvarus (adequate reduction quality). When the femoral neck angle of the operated hip was reduced compared with that of the contralateral hip, we defined it as varus alignment. The original authors have established a detailed definition of varus.3.Local alignment on the anteromedial cortex: We classified it as either intramedullary type or nonintramedullary type (adequate reduction quality) [13,14]. The intramedullary type was defined as a head–neck fragment located laterally or posteriorly to the cortex of the femoral shaft [14].

### 2.4. Screw Placement in the Femoral Head

4.TAD [12]. It was classified as TAD ≥ 25 or not TAD ≥ 25 (adequate screw placement). If the original authors used continuous values or the other TAD cut-off values, we requested a reanalysis with a cut off value of 25 mm from them [4,12].5.Screw placement in the femoral head [15]. It was classified as adequate or inadequate screw placement. Adequate screw placement was defined as center–center or inferior–center placement [16]. If the original authors used the Parker’s ratio [17], we requested them for a reanalysis.

### 2.5. Outcomes

The primary outcome of this study was failed internal fixation, which indicated complications with fracture healing following internal fixation. Failed internal fixation included cut-out, cut-through, nonunion, implant failure, and over sliding distance, all of which required reoperation. We considered any of these as a failed internal fixation. Cut-out indicated the screw cut-out in any direction from the femoral head.

### 2.6. Search Method

We searched the Cochrane central register of controlled trials, MEDLINE, Embase, and study registries including clinicaltrials.gov and international clinical trials registry platform on 27 November 2021, using the search strategy (Appendix A);. We also manually searched the reference lists of the included studies and international guidelines [18,19].

### 2.7. Study Selection and Data Extraction

Three pairs of reviewers independently screened every title and abstract of the articles based on the inclusion and exclusion criteria. Any conflicts between the two reviewers were resolved through discussion or with the help of a third reviewer.

Two reviewers on each trial independently performed data extraction of the included trials. For the effective measurement of the outcomes, we extracted both crude and multivariable estimates. We predefined the following possible confounders: age, sex, and fracture type (stable or unstable type based on AO/OTA classification (31A1, 2, 3)) [11,20]. If the possible confounders were not adjusted for in the original studies, we requested a re-analysis from the original authors.

### 2.8. Risk of Bias (Quality) Assessment

Two reviewers on each trial independently assessed the risk of bias using the quality in prognosis studies (QUIPS) tool [21]. If the possible confounders were adjusted for in the original studies, we rated the study as “low risk bias” for confounding factors. The overall risk of bias was rated low if all the QUIPS domains were rated low, high if two or more domains were with high risk of bias, and moderate in the other ratings.

### 2.9. Statistical Analysis

We pooled the crude and adjusted odds ratios (ORs) with 95% confidence intervals (CIs) for all outcomes using a random-effects meta-analysis model weighted by the inverse variance estimate. When both crude and adjusted ORs were used in the original papers, adjusted ORs were selected. We calculated the number needed to harm (NNH) for each outcome using the pooled relative risk and the median event rate of the control group.

We evaluated the statistical heterogeneity by visual inspection of the forest plots and calculation of the I^2^ statistic. We assessed the reason for heterogeneity when it was moderate or high (I^2^ > 50%). If more than 10 studies were included in each outcome, we performed a funnel plot analysis and the Egger’s test to assess reporting bias.

We judged the GRADE criteria for each association [9] (Table 1). We used the informative statements corresponding to the GRADE criteria [22].

A prespecified subgroup analysis was performed for the fracture type (stable or unstable) and the implant type used for internal fixation (SHS or CMN). The stable type was defined as A1 and A2.1 in the 2007 classification [20] and A1 in the 2018 classification [11] according to the AO/OTA guidelines.

We used crude ORs in the sensitivity analysis. As a post hoc analysis, we used adjusted and pooled ORs in studies or excluded studies with a high risk of bias in the overall risk of bias domain. 

All analyses were performed using RevMan 5.4 (Cochrane Collaboration, London, UK) and STATA 17.0 (Stata-Corp LP, College Station, TX, USD). Values of *p* < 0.05 were considered significant.

## 3. Results

After screening 4974 records, we included 36 studies (8938 participants) in the final analysis (Figure 1) [14,23,24,25,26,27,28,29,30,31,32,33,34,35,36,37,38,39,40,41,42,43,44,45,46,47,48,49,50,51,52,53,54]. Appendix A shows the excluded studies with reasons for their exclusion. By contacting the authors, we obtained unpublished data from 11 studies (Appendix A) [38,39,40,41,42,44,45,46,47,48]. The characteristics of the included studies are summarized in Appendix A.

In the 36 included studies, 31 were retrospective cohort studies, three were prospective cohort studies, and two were case-control studies. Twenty-four (66.7%) studies included participants from Asian countries. Eighteen studies (50.0%) reported both TAD and cut-out values.

Seven studies on failed internal fixation and 12 studies on cut-out had zero outcome events in at least one arm. Studies with no outcome events in both arms were excluded from the meta-analysis. A few studies had assessed the association with adjusted ORs. Appendix A presents the quality assessment results of the included studies using the QUIPS tool. The overall risk of bias in most of the included studies was moderate or high, mainly because of insufficient adjustment for confounders. The prevalence (median, interquartile range) of each postoperative radiographic findings was as follows: poor reduction based on the Baumgaertner criteria (5.9%, 4.0–10.9%), varus malreduction (3.3%, 3.1–17.1%), intramedullary malreduction on the anteromedial cortex (22.4%, 11.4–28.6%), TAD ≥ 25 (17.5%, 6.8–25.5%), and inadequate screw placement in the femoral head (44.3%, 30.0–56.0%). The incidence rate (median, interquartile range) of each outcome was as follows: failed internal fixation (2.8%, 0–7.0%) and cut-out (5.6%, 0.5–8.1%).

### 3.1. Association between Immediate Postoperative Radiographic Findings and Failed Internal Fixations

The findings of this review are summarized in Table 1 (Appendix A). Moderate certainty of evidence supported that intramedullary malreduction on the anteromedial cortex (pooled OR 7.23, 95% CI 2.49–21.01; NNH 21, 95% CI 7–83) (Appendix A) was associated with failed internal fixation. 

The low certainty of evidence supported that the association with failed internal fixation was ranked using pooled ORs as follows: inadequate screw placement in the femoral head (pooled OR 6.11, 95% CI 1.00–37.27; NNH 6, 95% CI 2–12151) (Appendix A) and varus malreduction (pooled OR 4.48; 95% CI 1.03–19.46; NNH 8, 95% CI 2–1000) (Appendix A) were associated with failed internal fixation. The evidence on whether TAD ≥ 25 (pooled OR 1.39, 95% CI 0.65–2.98; NNH 200, 95% CI −200–37) (Appendix A) and poor reduction based on the Baumgaertner criteria is associated with failed internal fixation (pooled OR 1.17, 95% CI 0.01–92.75; NNH 500, 95% CI −71–2) (Appendix A) is uncertain.

### 3.2. Association between Immediate Postoperative Radiographic Findings and Cut-Outs

Low certainty of evidence supported that the association with cut-out was ranked using pooled ORs as follows: TAD ≥ 25 (pooled OR 7.92, 95% CI 3.33–18.82; NNH 14, 95% CI 6–40) (Appendix A) and poor reduction by Baumgaertner criteria (pooled OR 7.78, 95% CI 3.41–17.76; NNH 5, 95% CI 3–11) (Appendix A), poor reduction by intramedullary malreduction on the anteromedial cortex (pooled OR 7.35, 95% CI 2.83–19.10; NNH 53, 95% CI 20–200) (Appendix A) , varus malreduction (pooled OR 4.54, 95% CI 2.17–9.49; NNH 28, 95% CI 11–200) (Appendix A). The evidence on whether inadequate screw placement in the femoral head (pooled OR 2.75, 95% CI 1.03–7.33; NNH 16, 95% CI 5–1000) (Appendix A) is associated with cut-out is uncertain.

Appendix A depicts study estimates in a funnel plot. The Egger’s test results for the funnel plot asymmetry were insignificant; however, the plot for the association between poor reduction by Baumgaertner criteria or inadequate screw placement in the femoral head and cut-out was asymmetrical (*p* value = 0.15, 0.17, respectively).

### 3.3. Subgroup and Sensitivity Analyses

No study had assessed the association in only participants with stable fracture types (Appendix A). The subgroup analysis indicated a significant difference between implant types (SHS and CMN) in the association between poor reduction based on the Baumgaertner criteria and cut-out *(p* = 0.008 for the interaction), between TAD ≥ 25 and cut-out (*p* = 0.012 for the interaction), and between inadequate screw placement in the femoral head and cut-out *(p* < 0.001 for the interaction) (Table 2, Appendix A).

The sensitivity analyses performed by repeating the main analysis with crude ORs and adjusted ORs or excluding studies with a high risk of bias indicated a consistent association between intramedullary malreduction on the anteromedial cortex and failed internal fixation and between TAD ≥ 25 and cut-out (Table 3).

## 4. Discussion

The main findings of this systematic review and meta-analysis were that the evidence for most immediate postoperative radiographic findings as the prognostic factors for failed internal fixation is uncertain. Intramedullary malreduction on the anteromedial cortex is probably associated with an increased risk of failed internal fixation. TAD ≥ 25 may be associated with an increased risk of cut-out. The results were consistent with those of the sensitivity analyses; however, the robustness was uncertain owing to limited data in the subgroup analyses, including implant and fracture types.

In terms of pooled ORs, the intramedullary malreduction on the anteromedial cortex had the greatest impact on failed internal fixation. Based on the NNH 21 of intramedullary malreduction and its relatively high prevalence (22.4%) among postoperative radiographic findings, surgeons may first consider it as a prognostic factor for failed internal fixation. The findings are consistent with those of a previous review that discovered a link between intramedullary malreduction on the anteromedial cortex and nonunion [6]. We confirmed this association by supplementing the comprehensive review and GRADE assessment.

The usefulness of other postoperative radiographic findings as prognostic factors for failed internal fixation and cut-out is debatable. TAD ≥ 25 is recognized as the predictive factor based on consensus [4,5]; however, we demonstrated that the prognostic value of TAD ≥ 25 was uncertain based on the serious risk of bias and inconsistency in the results from the included studies. Moreover, other postoperative radiographic findings had a serious risk of bias and concerns about publication bias or imprecision, which had not been sufficiently assessed in earlier reviews [4,5]. Hence, routine assessment of these postoperative radiographic findings may be unnecessary because of lack of evidence. In the future, larger studies with adjustments for important confounders are warranted to determine which radiographic findings are clinically useful as prognostic factors. 

Our subgroup analysis revealed that the strength of the association between postoperative radiographic findings and prognosis may vary with implant types. However, these results should be interpreted with caution because of the small number of studies in each subgroup.

### 4.1. Limitations

This review has several limitations. First, despite the comprehensive search, the number of included original studies for each association was few (median, 5; interquartile range 4–6) (Table 1). In addition, we could not perform several prespecified subgroup and sensitivity analyses due to the limited data. 

Second, we included some studies with zero outcome events in either arm owing to the small sample size. Hence, the incidence of failed internal fixation was lower than that of a previous report [55]. Third, this review only indicated the certainty of evidence on prognostic factor estimates and not causal relationships. Further prospective studies should be performed to investigate the effectiveness of modifying these prognostic factors to determine the causal relationships. Finally, the results may not be generalizable to other races because most of the included studies were conducted in Asia.

### 4.2. Strengths

Despite the limitations mentioned above, this is the first review to provide comprehensive evidence on the association between postoperative findings and failed internal fixation, which compensates for the scarcity of evidence in earlier systematic reviews [4,5,6]. Our findings are useful for establishing future clinical guidelines, intraoperative decision-making, postoperative follow-up management, and constructing clinical risk prediction models for failed internal fixation. Moreover, this review was based on rigorous methodology such as the MOOSE guidelines, Cochrane handbook, and GRADE recommendations [7,8,9]. Finally, this review includes unpublished data and adjusted ORs following a reanalysis to adjust for possible confounders.

## 5. Conclusions

Most postoperative radiographic findings on immediate postoperative radiographs for trochanteric fractures were uncertain as prognostic factors for failed internal fixations. For internally fixed trochanteric fractures, surgeons may consider the intramedullary malreduction on the anteromedial cortex to estimate the risk of failed internal fixation. However, we may need to reassess whether postoperative radiographic findings are a reliable prognostic factor with further rigid research because the other postoperative radiographic findings have little or no evidence. Further well-designed studies considering a larger sample size and adjustment for confounders would improve the certainty of the evidence.

## Figures and Tables

**Figure 1 jcm-11-04879-f001:**
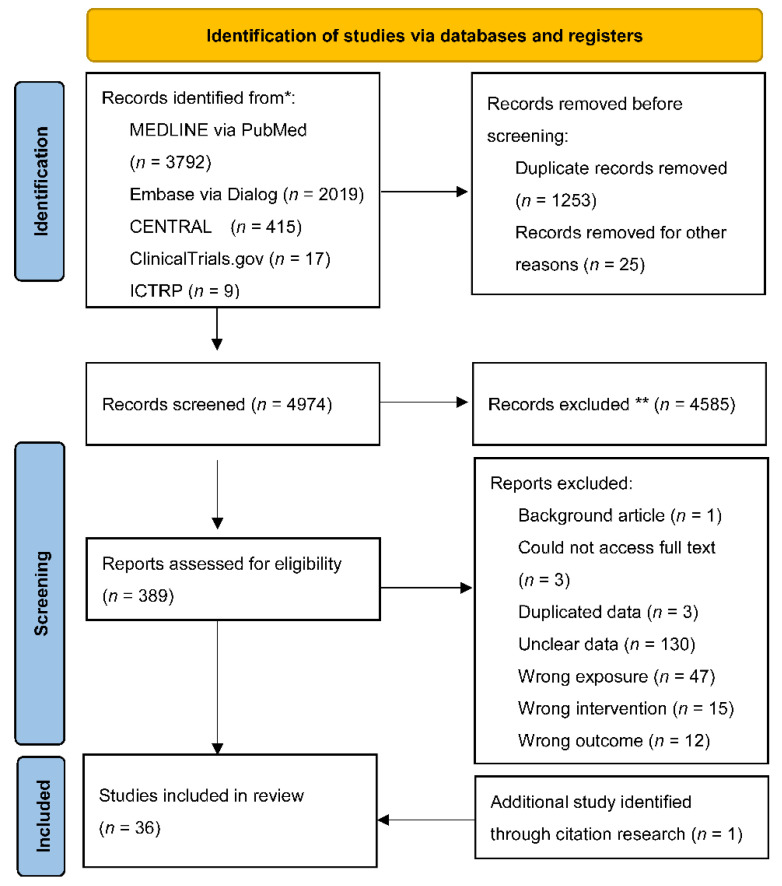
Preferred reporting items for systematic reviews and meta-analysis (PRISMA) flowchart. International Clinical Trials Registry Platform (ICTRP). * Consider, if feasible to do so, reporting the number of records identified from each database or register searched (rather than the total number across all databases/registers). ** If automation tools were used, indicate how many records were excluded by a human and how many were excluded by automation tools.

**Table 1 jcm-11-04879-t001:** Summary of findings on association of immediate postoperative radiographic findings and failed internal fixations in patients treated with internal fixation for trochanteric fractures.

Predictive Factors	Outcomes	Anticipated Absolute Effects ^a^ (95% CI)		Relative Effect (95% CI)	No. of Participants (Observational Studies)	Certainty of Evidence (Grade) ^c^
		Assumed Risk with Comparator ^b^	Corresponding Risk with Predictive Factor			
Poor reduction by Baumgaertner criteria	Failed internal fixation	14 per 1000	16 per 1000 (0–568)	OR 1.17 (0.01–92.75)	427 (2 studies)	Very low ^d,e,f,g^
	Cut-out	42 per 1000	255 per 1000 (130–439)	OR 7.78 (3.41–17.76)	3250 (9 studies)	Low ^d,h^
Varus malreduction	Failed internal fixation	45 per 1000	174 per 1000 (46–478)	OR 4.48 (1.03–19.46)	832 (2 studies)	Low ^f,g^
	Cut-out	25 per 1000	61 per 1000 (30–119)	OR 4.54 (2.17–9.49)	1194 (5 studies)	Low ^d,g^
Intramedullary malreduction on anteromedial cortex	Failed internal fixation	8 per 1000	55 per 1000 (20–145)	OR 7.23 (2.49–21.01)	1146 (4 studies)	Moderate ^d^
	Cut-out	3 per 1000	22 per 1000 (8–54)	OR 7.35 (2.83–19.10)	1383 (6 studies)	Low ^d,g^
TAD ≥ 25	Failed internal fixation	14 per 1000	19 per 1000 (9–41)	OR 1.39 (0.65–2.98)	1013 (4 studies)	Very low ^d,e,g^
	Cut-out	11 per 1000	81 per 1000 (36–173)	OR 7.92 (3.33–18.82)	4196 (13 studies)	Low ^d,e^
Inadequate screw placement in femoral head	Failed internal fixation	43 per 1000	215 per 1000 (43–626)	OR 6.11 (1.00–37.27)	998 (5 studies)	Low ^d,g^
	Cut-out	41 per 1000	105 per 1000 (42–239)	OR 2.75 (1.03–7.33)	2399 (9 studies)	Very low ^d,e,g,h^

CI, confidence interval; OR, odds ratio; TAD, tip–apex distance. ^a^ Risk in the exposed group (and its 95% confidence interval) was based on the assumed risk in the nonexposed group and the relative effect of the exposure (and its 95% CI). ^b^ Median event rate of the included studies. ^c^ GRADE Working Group grades of evidence. ^d^ Downgraded by one level because of serious risk of bias. ^e^ Downgraded by one level because of inconsistency. ^f^ Downgraded by one level because the number of participants in the exposed group was small. ^g^ Downgraded by one level because of imprecision. ^h^ Downgraded by one level because of publication bias.

**Table 2 jcm-11-04879-t002:** Subgroup analysis on the association between immediate postoperative radiographic findings and failed internal fixations by implant type.

		SHS			CMN			
Prognostic Factors	Outcomes	No of Studies	I^2^ (%)	Pooled Odds Ratio (95% CI)	No of Studies	I^2^ (%)	Pooled Odds Ratio (95% CI)	*p* for Interaction
Poor reduction by Baumgaertner criteria	Failed internal fixation	1	–	7.72 (1.85–32.23)	1	–	9.46 (2.36–37.83)	–
	Cut-out	1	–	1.99 (1.05–3.79)	4	48	10.47 (3.68–29.79)	0.008
Varus malreduction	Failed internal fixation	1	–	4.60 (1.02–20.85)	1	–	2.89 (0.006–1418.51)	–
	Cut-out	1	–	4.38 (1.34–14.24)	3	0	3.59 (1.25–10.33)	0.807
Intramedullary malreduction	Failed internal fixation	–	–	–	4	0	6.73 (2.42–18.71)	–
	Cut-out	–	–	–	6	0	7.35 (2.83–19.10)	–
TAD ≥ 25	Failed internal fixation	1	–	0.71 (0.21–2.41)	3	0	2.13 (0.81–5.60)	0.168
	Cut-out	1	–	24.56 (12.52–48.18)	9	62	4.68 (1.55–14.09)	0.012
Inadequate screw placement in femoral head	Failed internal fixation	1	–	1.53 (0.53–4.44)	4	72	11.39 (0.83–156.91)	0.165
	Cut-out	1	–	0.36 (0.22–0.59)	8	2.9	3.30 (1.96–5.56)	<0.001

SHS, sliding hip screw; CMN, cephalomedullary nail; CI, confidence interval; TAD, tip–apex distance.

**Table 3 jcm-11-04879-t003:** Sensitivity analysis on association between immediate postoperative radiographic findings and failed internal fixations.

		Crude Odds Ratios			Adjusted Odds Ratios			Repeating the Main Analysis Excluding Studies with High Risk of Bias ^a^		
Prognostic Factors	Outcomes	No of Studies	I^2^ (%)	Pooled Odds Ratio (95% CI)	No of Studies	I^2^ (%)	Pooled Odds Ratio (95% CI)	No of Studies	I^2^ (%)	Pooled Odds Ratio (95% CI)
Poor reduction by Baumgaertner criteria	Failed internal fixation	2	0	8.57 (3.17–23.18)	2	91.3	1.17 (0.02–92.75)	2	91.3	1.17 (0.01–92.75)
	Cut-out	9	77.6	7.56 (3.18–18.00)	4	47	11.18 (3.46–36.16)	9	68	7.78 (3.41–17.76)
Varus malreduction	Failed internal fixation	2	0	5.45 (1.86–15.92)	2	0	4.48 (1.03–19.46)	2	0	4.48 (1.03–19.46)
	Cut-out	5	0	4.60 (2.31–9.17)	2	0	5.66 (1.98–16.20)	3	0	5.17 (1.92–13.90)
Intramedullary malreduction	Failed internal fixation	4	0	7.56 (2.80–20.45)	1	–	9.58 (0.88–104.55)	4	0	6.73 (2.42–18.71)
	Cut-out	6	0	8.81 (3.12–21.11)	1	–	7.00 (0.81–60.89)	6	0	7.35 (2.83–19.10)
TAD ≥ 25	Failed internal fixation	4	0	1.42 (0.71–2.85)	1	-	1.52 (0.21–11.08)	3	0	1.07 (0.40–2.88)
	Cut-out	13	77.5	5.64 (2.50–12.72)	2	0	20.81 (6.09–71.12)	9	73.4	6.92 (2.30–20.78)
Inadequate screw placement in femoral head	Failed internal fixation	5	74.8	5.52 (1.71–17.81)	2	0	4.86 (0.04–562.58)	4	0	2.00 (0.91–4.37)
	Cut-out	9	84.3	3.01 (1.17–7.77)	2	13	2.64 (0.78–8.94)	7	85.6	3.15 (0.90–11.04)

CI, confidence interval; TAD, tip–apex distance. ^a^ When both crude and adjusted odds ratios were used in the original paper, adjusted odds ratio was selected.

## Data Availability

The data presented in this study are available on request from the corresponding author.

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
