# Peer review of "Association between Immediate Postoperative Radiographic Findings and Failed Internal Fixation for Trochanteric Fractures: Systematic Review and Meta-Analysis"

_jcm, 2022, doi:10.3390/jcm11164879_

Round 1

Reviewer 1 Report

Manuscript ID: jcm-1830678

The authors performed rigorous statistical analysis to evaluate association between immediate postoperative radiologic findings and internal-fixation failure. However, the following limitations should be warranted to be published as an original study.

1. The major concern of current study is that the authors tried to ‘meta-analyze’ too many associations, without consideration of the number of studies included in their meta-analysis. Actually, substantial proportions of their results are based on 2 to 4 studies, most of which came from China or Japan. Therefore, their results are strongly influenced by publication bias.

2. In addition, their results, ‘Most postoperative radiographic findings on immediate postoperative radiographs were uncertain as a prognostic factor for failed internal fixation and cut-out’, cannot be easily accepted or be persuasive to orthopedic surgeon.

Minor comments>

1. Please revise the title clarifying this study is only focused on trochanter fracture.

2. Page 7 line 228: Please correct the figure citation: Appendix B à Appendix C

3. In the results section, please clarify the included studies to estimate pooled odds ratios by citing the corresponding figures for all results

Author Response

Response to Reviewer 1 Comments:

The authors performed rigorous statistical analysis to evaluate association between immediate postoperative radiologic findings and internal-fixation failure. However, the following limitations should be warranted to be published as an original study.

Response

We thank the Reviewer 1 very much for these comments. We have revised the related text according to the comments.

  1. The major concern of current study is that the authors tried to ‘meta-analyze’ too many associations, without consideration of the number of studies included in their meta-analysis. Actually, substantial proportions of their results are based on 2 to 4 studies, most of which came from China or Japan. Therefore, their results are strongly influenced by publication bias.

Response

Thank you for this valuable comment. To our knowledge, the number of studies included in a meta-analysis does not influence publication bias [1]. As previously reported, what is important is not the number of studies, but whether the search is comprehensive and seeks unpublished data [2].

We endeavored to include all available studies using a comprehensive search, including contact with authors according to the MOOSE checklist. As a result, we obtained unpublished data from 11 studies (Appendix B, Table 2). Although the small number of included studies did not allow us to perform a funnel plot analysis and Egger's test to assess reporting bias, this is often the case in systematic review [3]. We have added the following sentences as a further study limitation.

Page 12:

First, despite the comprehensive search, the number of included original studies for each association was few (median, 5; interquartile range 4-6) (Table 1). In addition, we could not perform several pre-specified subgroup and sensitivity analyses due to the limited data. 

[1] Cochrnae handbook; https://training.cochrane.org/handbook/archive/v6.2/chapter-07

[2] Ziai H, Zhang R, Chan AW, Persaud N. Search for unpublished data by systematic reviewers: an audit. BMJ Open. 2017;7(10):e017737.

[3] Page MJ, Shamseer L, Altman DG, et al. Epidemiology and Reporting Characteristics of Systematic Reviews of Biomedical Research: A Cross-Sectional Study. PLoS Med. 2016;13(5):e1002028.

  1. In addition, their results, ‘Most postoperative radiographic findings on immediate postoperative radiographs were uncertain as a prognostic factor for failed internal fixation and cut-out’, cannot be easily accepted or be persuasive to orthopedic surgeon.

Response

Thank you for this suggestion. As the reviewer pointed out, the interpretation of GRADE may not be familiar with some readers. We used the informative statements corresponding to the GRADE criteria [22]. The GRADE informative statement recommends that authors state the following in line with the certainty of evidence. 

LOW Certainty of the evidence

The informative statements should be based on both size of the effect and certainty of evidence. The authors selected the qualifying words “may, appears, suggests, and likely“ as informative statements  .

VERY LOW Certainty of the evidence

The informative statement is the same sentence without considering “effect size”; The evidence is “very uncertain” about the effect of the prognostic factor on the outcome.

   Our assessment of the certainty of evidence was moderate certainty in one association, low in six associations, and very low in three associations as per the assessment of GRADE for a total of ten associations. Therefore, we interpreted that most associations were uncertain of evidence in this study. We have edited the sentences as follows.

Page 5:

We judged the GRADE criteria for each association [9] (Table 1). We used the informative statements corresponding to the GRADE criteria [22].

Abstract, conclusions:

Most postoperative radiographic findings on immediate postoperative radiographs for trochanteric fractures were uncertain as prognostic factors for failed internal fixations. 

[9] Foroutan, F.; Guyatt, G.; Zuk, V.; Vandvik, P.O.; Alba, A.C.; Mustafa, R.; Vernooij, R.; Arevalo-Rodriguez, I.; Munn, Z.; Roshanov, P.; Riley, R.; Schandelmaier, S.; Kuijpers, T.; Siemieniuk, R.; Canelo-Aybar, C.; Schunemann, H.; Iorio, A. GRADE Guidelines 28: Use of GRADE for the assessment of evidence about prognostic factors: rating certainty in identification of groups of patients with different absolute risks. J Clin Epidemiol 2020, 121, 62-70. DOI:10.1016/j.jclinepi.2019.12.023.

[22] Santesso N, Glenton C, Dahm P, Garner P, Akl EA, Alper B, Brignardello-Petersen R, Carrasco-Labra A, De Beer H, Hultcrantz M, Kuijpers T, Meerpohl J, Morgan R, Mustafa R, Skoetz N, Sultan S, Wiysonge C, Guyatt G, Schünemann HJ; GRADE Working Group. GRADE guidelines 26: informative statements to communicate the findings of systematic reviews of interventions. J Clin Epidemiol. 2020, 119, 126-135. DOI: 10.1016/j.jclinepi.2019.10.014. Epub 2019.

Minor comments>

  1. Please revise the title clarifying this study is only focused on trochanter fracture.

Response

As the reviewer correctly pointed out, we have revised the title.

Title:

Association between immediate postoperative radiographic findings and failed internal fixation for trochanteric fractures: systematic review and meta-analysis

  1. Page 7 line 228: Please correct the figure citation: Appendix B à Appendix C

Response

As the reviewer correctly pointed out, we have revised the sentence.

Page 7:  

The findings of this review are summarized in Table 1 (Appendix C Figures 1–10).

  1. In the results section, please clarify the included studies to estimate pooled odds ratios by citing the corresponding figures for all results

Response

As the reviewer pointed out, we added citing the corresponding figures as follows.

Page 7:

Moderate certainty of evidence supported that intramedullary malreduction on the anteromedial cortex (pooled OR 7.23, 95% CI 2.49–21.01; NNH 21, 95% CI 7–83) (Appendix C Figure 5) was associated with failed internal fixation. 

Page 8:

The low certainty of evidence supported that the association with failed internal fixation was ranked using pooled ORs as follows: inadequate screw placement in the femoral head (pooled OR 6.11, 95% CI 1.00–37.27; NNH 6, 95% CI 2–12151) (Appendix C Figure 9) and varus malreduction (pooled OR 4.48; 95% CI 1.03–19.46; NNH 8, 95% CI 2–1000)  (Appendix C Figure 3) were associated with failed internal fixation. The evidence on whether TAD ≥ 25 (pooled OR 1.39, 95% CI 0.65–2.98; NNH 200, 95% CI -200–37)  (Appendix C Figure 7) and poor reduction based on the Baumgaertner criteria is associated with failed internal fixation (pooled OR 1.17, 95% CI 0.01–92.75; NNH 500, 95% CI -71–2) (Appendix C Figure 1) is uncertain.

Low certainty of evidence supported that the association with cut-out was ranked using pooled ORs as follows: TAD ≥ 25 (pooled OR 7.92, 95% CI 3.33–18.82; NNH 14, 95% CI 6–40) (Appendix C Figure 8) and poor reduction by Baumgaertner criteria (pooled OR 7.78, 95% CI 3.41–17.76; NNH 5, 95% CI 3–11) (Appendix C Figure 2) , poor reduction by intramedullary malreduction on the anteromedial cortex (pooled OR 7.35, 95% CI 2.83–19.10; NNH 53, 95% CI 20–200) (Appendix C Figure 6) , varus malreduction (pooled OR 4.54, 95% CI 2.17–9.49; NNH 28, 95% CI 11–200)  (Appendix C Figure 4). The evidence on whether inadequate screw placement in the femoral head (pooled OR 2.75, 95% CI 1.03–7.33; NNH 16, 95% CI 5–1000) (Appendix C Figure 10) is associated with cut-out is uncertain.

Reviewer 2 Report

Thank you for the opportunity to revise this review article, focusing on the relationship between specific post-operative radiographic findings and failure of internal fixation in trochanteric fractures.

The topic is always very relevant. The study is well conducted and the article well written. I particularly congratulate the authors for the depth of the analysis conducted. 

The introduction is brief but comprehensive, the methodology well presented, the results are clear, the discussion covers the key points, and the conclusions are consistent with the results. Although no groundbreaking data emerge from this study, the extent of the analysis performed and the methodology, make the article worthy of publication in my opinion. I only have one consideration for the authors: the title is not clear, the site of the fractures analyzed should be specified.

Thank you.

Author Response

Response to Reviewer 2 Comments:

Thank you for the opportunity to revise this review article, focusing on the relationship between specific post-operative radiographic findings and failure of internal fixation in trochanteric fractures.

The topic is always very relevant. The study is well conducted and the article well written. I particularly congratulate the authors for the depth of the analysis conducted. 

The introduction is brief but comprehensive, the methodology well presented, the results are clear, the discussion covers the key points, and the conclusions are consistent with the results. Although no groundbreaking data emerge from this study, the extent of the analysis performed and the methodology, make the article worthy of publication in my opinion. I only have one consideration for the authors: the title is not clear, the site of the fractures analyzed should be specified.

Thank you. 

Response

We thank the Reviewer 2 very much for these comments. As the reviewer pointed out, we have revised the title.

Title:

Association between immediate postoperative radiographic findings and failed internal fixation for trochanteric fractures: systematic review and meta-analysis